# The Overactivation of NADPH Oxidase during *Clonorchis sinensis* Infection and the Exposure to *N*-Nitroso Compounds Promote Periductal Fibrosis

**DOI:** 10.3390/antiox10060869

**Published:** 2021-05-28

**Authors:** Ji Hoon Jeong, Junyeong Yi, Myung Ki Hwang, Sung-Jong Hong, Woon-Mok Sohn, Tong-Soo Kim, Jhang Ho Pak

**Affiliations:** 1Department of Convergence Medicine, University of Ulsan College of Medicine and Asan Institute for Life Sciences, Asan Medical Center, 88 Olympic-ro 43-gil, Songpa-gu, Seoul 05505, Korea; otooo10@mail.ulsan.ac.kr (J.H.J.); junyi@mail.ulsan.ac.kr (J.Y.); 2Department of Tropical Medicine and Parasitology, School of Medicine, Inha University, 100 Inha-ro, Michuhol-gu, Incheon 22212, Korea; hmk4688@inha.ac.kr (M.K.H.); tongsookim@inha.ac.kr (T.-S.K.); 3Department of Medical Environmental Biology, College of Medicine, Chung-Ang University, 84 Heuksuk-ro, Dongjak-gu, Seoul 06974, Korea; hongsj@cau.ac.kr; 4Department of Parasitology and Tropical Medicine, Institute of Health Sciences, College of Medicine, Gyeongsang National University, 79 Gangnam-ro, Jinju 52727, Korea; wmsohn@gnu.ac.kr

**Keywords:** *Clonorchis sinensis* infection, excretory-secretory products, *N*-nitrosodimethylamine, cholangiocytes, oxidative stress, NADPH oxidases, periductal fibrosis, cholangiocarcinoma

## Abstract

*Clonorchis sinensis*, a high-risk pathogenic human liver fluke, provokes various hepatobiliary complications, including epithelial hyperplasia, inflammation, periductal fibrosis, and even cholangiocarcinogenesis via direct contact with worms and their excretory–secretory products (ESPs). These pathological changes are strongly associated with persistent increases in free radical accumulation, leading to oxidative stress-mediated lesions. The present study investigated *C. sinensis* infection- and/or carcinogen *N*-nitrosodimethylamine (NDMA)-associated fibrosis in cell culture and animal models. The treatment of human cholangiocytes (H69 cells) with ESPs or/and NDMA increased reactive oxidative species (ROS) generation via the activation of NADPH oxidase (NOX), resulting in augmented expression of fibrosis-related proteins. These increased expressions were markedly attenuated by preincubation with a NOX inhibitor (diphenyleneiodonium chloride) or an antioxidant (*N*-acetylcysteine), indicating the involvement of excessive NOX-dependent ROS formation in periductal fibrosis. The immunoreactive NOX subunits, p47^phox^ and p67^phox^, were observed in the livers of mice infected with *C. sinensis* and both infection plus NDMA, concomitant with collagen deposition and immunoreactive fibronectin elevation. Staining intensities are proportional to lesion severity and infection duration or/and NDMA administration. Thus, excessive ROS formation via NOX overactivation is a detrimental factor for fibrogenesis during liver fluke infection and exposure to N-nitroso compounds.

## 1. Introduction

Oxidative stress results from an excessive pro-oxidant formation to antioxidant detoxifying capacity, exerting oxidative damages in tissues and organs. Free radicals (reactive oxygen and nitrogen species; ROS/RNS) are generated in the liver as byproducts during the metabolizing processes of various compounds as well as a mitochondrial electron chain reaction. Moreover, their levels are increased by ischemia/reperfusion; alcohol overconsumption; xenobiotics or heavy metals intoxications; and bacterial, viral, and parasitic infections [1,2]. Aberrant accumulations of these free radicals contribute to the initiation of inflammation and subsequent development of fibrosis, leading to more severe hepatic complications.

Clonorchiasis, an infectious disease caused by an oriental liver fluke, *Clonorchis sinensis*, is endemic in Southeast Asian countries, constituting a significant public health concern [3]. These flukes reside in peripheral bile ducts and provoke various pathological changes in the bile ducts and surrounding liver tissues. The early stage of infection results in biliary epithelial proliferation and inflammation around the biliary trees. The epithelial hyperplasia at later stages proceeds to adenomatous hyperplasia and glandular proliferation in the biliary epithelia with facultative goblet cell accumulation. The immune responses transform into prolonged inflammation with extensive fibrotic deposition along the bile duct wall, which creates a vulnerable environment for the development of advanced hepatobiliary diseases including cholangiocarcinogenesis at the chronic infection stage [4,5]. The International Agency for Research on Cancer has classified *C. sinensis* as a definite biological human carcinogen as previously done with *Opisthorchis viverrini*, a cholangiocarcinoma (CCA) promoter in endemic regions [6]. Chronic hepatobiliary injury associated with liver fluke infection, a precursor to CCA, is a multifactorial outcome of mechanical and biochemical irritations of biliary epithelia caused by the fluke sucking activity, metabolites, excretory–secretory products (ESPs), and subsequent secondary bacterial infection. These combined actions exert host immunopathological responses (e.g., prolonged inflammation, increases in free radicals, and advanced periductal fibrosis) leading to the accumulation of oxidative/nitrative DNA lesions and neoplasms [7,8]. This intrahepatic neoplastic lesion leads to mass-forming CCA in Syrian golden hamsters when administered with a subcarcinogenic dose of *N*-nitrosodimethylamine (NDMA), a potent carcinogen and hepatotoxin [9,10].

Fibrosis is the pathological consequence of repetitive tissue injuries and dysregulated wound-healing processes, along with chronic inflammation reactions, resulting in excessive deposition and altered composition of extracellular matrix (ECM). Fibrogenesis is prominently mediated by activated fibroblasts or myofibroblasts, which are responsible for matrix remodeling and increased production of ECM proteins such as collagen, fibronectin, laminin, and other fewer elements [11]. Periductal fibrosis in the livers of mice infected with *C. sinensis* increased in proportion to infective periods, concomitant with inflammation and appearance of oxidative markers including lipid peroxidation and oxidative DNA adducts [12]. In addition, the formation of the fibrous stroma of CCA tissues with dense collagen fiber deposition was observed around the bile ducts of *C. sinensis*-associated CCA hamster livers [13]. Periductal fibrosis accompanied with collagen I and III accumulation were increased in a time-dependent manner in the livers of hamsters infected with *O. viverrini*, wherein the activations of metalloproteinase (MMP) isozymes contributed to both early inflammation reactions and late fibrogenesis [14].

NADPH oxidases (NOXs) are multicomponent enzymes composed of two intrinsic membrane subunits (gp22^phox^ and gp91^phox^) and at least four cytosolic subunits (p40^phox^, p47^phox^, p67^phox^, and GTPase Rac) [15]. The cytosolic components on activation translocate to the plasma membrane and associate with the membrane-bound subunits to generate superoxide anion with subsequent dismutation to hydrogen peroxide during the catalytic metabolism of molecular oxygen for a variety of redox signaling and host defense [2]. Seven isoforms of NOX have been identified so far in mammalian cells, including NOX1-5, DUOX1, and DUOX2. In the liver, hepatocytes and hepatic stellate cells (HSCs) express NOX1, NOX2, NOX4, DUOX1, and DUOX2; endothelial cells express NOX1, NOX2, and NOX4; and Kupffer cells express phagocytic NOX2. Each NOX isoform knockout mice have shown decreased liver inflammation and fibrosis induced by CCl_4_ injection or bile duct ligation, indicating that NOXs play essential roles in the development and progression of hepatic apoptosis and fibrosis [16]. ROS generation via NOX activation was previously reported to promote nuclear factor-κB (NF-κB)-mediated inflammation in CCA cells (HuCCT1) exposed to *C. sinensis* ESPs [17]. Moreover, NOX activation induced by ESPs was attenuated by antioxidant enzyme overexpression [18]. However, its direct involvement in biliary fibrosis associated with clonorchiasis remains to be elucidated.

Dysregulated ROS generation is an important etiological factor for the initiation of hepatobiliary disorders associated with chronic liver fluke infection (e.g., inflammation and periductal fibrosis). Thus, this study examined NOX activation and the expression of fibrosis-related proteins in human cholangiocytes (H69 cells) treated with *C. sinensis* ESPs and NDMA. Moreover, this study also aimed to determine the differential immunoreactive expression of NOX subunits and fibronectin and collagen deposition in the livers of infected mice according to NDMA administration and *C. sinensis* infection duration.

## 2. Materials and Methods

### 2.1. Materials

Cell culture medium components were purchased from Life Technologies (Grand Island, NY, USA) unless otherwise indicated. NDMA and *N*-acetylcysteine (NAC) were obtained from Sigma-Aldrich (St. Louis, MO, USA). Diphenyleneiodonium chloride (DPI) was obtained from Calbiochem (La Jolla, CA, USA). Polyclonal antibodies against the following proteins were obtained from the indicated sources: p47^phox^ (610354) and p67^phox^ (610912) (BD Biosciences, San Jose, CA, USA), calnexin (sc-23954, Santa Cruz Biotechnology, Santa Cruz, CA, USA), fibronectin (ab24136) and collagen I (ab6308, Abcam, Cambridge, UK), and glyceraldehyde-3-phosphate dehydrogenase (GAPDH; A300-641A, Bethyl Laboratories, Inc., Montgomery, TX, USA). Horseradish peroxidase (HRP)-conjugated secondary antibodies were purchased from Bethyl Laboratories. All other chemicals (biotechnology grade) were purchased from Sigma-Aldrich.

### 2.2. Preparation of C. sinensis ESPs

*C. sinensis* ESPs were prepared as previously described [17]. In brief, adult worms were recovered from bile ducts of New Zealand albino rabbits infected with ~500 metacercariae for 12 weeks. They were then washed several times with cold phosphate-buffered saline (PBS) to remove any host contaminants. Five fresh worms were cultured in 1 mL of prewarmed PBS containing antibiotic mixture and protease inhibitor cocktail (Sigma-Aldrich) for 3 h at 37 °C in a 5% CO_2_ incubator. The culture fluid was then pooled, centrifuged, concentrated with a Centriprep YM-10 filter unit (Merck Millipore, Billerica, MA, USA), and filtered through a sterile 20 μm syringe membrane. After measuring the ESP protein concentration, ESP aliquots were stored at −80 °C until use.

### 2.3. Cell Culture and ESPs/NDMA Treatment

H69 cells, a SV40-transformed normal human cholangiocyte line, were derived from non-cancerous human liver [19]. H69 cells were cultured in Dulbecco’s modified Eagle’s medium (DMEM) and DMEM/F12 containing 10% fetal bovine serum (FBS), 100 μg/mL streptomycin, 100 U/mL penicillin, 5 μg/mL of insulin, 2.0 × 10^−9^ M triiodothyronine, 5 μg/mL of transferrin, 5.5 μM epinephrine, 1.1 μM hydrocortisone, 1.8 × 10^−4^ M adenine, and 1.6 μM EGF at 37 °C in a humidified 5% CO_2_ atmosphere. Cells were seeded on 60 mm culture dishes and cultured for 24 h under standard conditions for ESPs, NDMA, or ESPs plus NDMA treatments. Cells were incubated in culture medium containing 2% FBS overnight, and then incubated for 3 h in serum-free medium. These serum-starved cells were treated with 1.6 μg/mL ESPs, 8 μg/mL NDMA, or ESPs plus NDMA for 24 h. Cells were pretreated with 1 mM NAC or 20 μM DPI for 1 h for the inhibition experiment. The culture media were then replaced with fresh media containing ESPs, NDMA, or ESPs plus NDMA and incubated as described above.

### 2.4. Ethics Statement 

Male FVB/NJ mice at 5–6 weeks old were purchased from Central Lab Animal Inc. (Seoul, Korea). The animal experiment protocol was reviewed and approved by the institutional animal care and use committee (IACUC) of Inha University (approval no. INHA161208-460) accredited with the Association for Assessment and Accreditation of Laboratory Animal Care (AAALAC). The animals were cared for and handled in the Inha University Animal Facility following the National Animal Care Policies (Accredited Unit, Korea FDA; unit number 36).

### 2.5. Preparation of C. sinensis Metacercariae and Infection

The metacercariae of *C. sinensis* were collected from naturally infected freshwater fish (*Pseudorashora parva*) in Korea. Fish were ground and digested in artificial gastric juice (0.4% pepsin, pH 1.0; MP Biochemicals Co., Solon, OH, USA). The coarse matter was removed from the digested content through a 1 mm mesh diameter sieve. The filtrate was allowed to settle and was washed several times by discarding the upper half and adding 0.85% saline. The metacercariae were collected under a dissecting microscope and stored in PBS containing antibiotic and antimycotics (Life Technologies) at 4 °C until use. Male FVB/NJ mice at 5–6 weeks old were randomly divided into four groups including 12 animals each. Control (the uninfected control), Cs, NDMA, and Cs plus NDMA groups were orally infected with 30 *C. sinensis* metacercariae, drank NDMA mixed with water, and received both 30 *C. sinensis* metacercariae and NDMA mixed with water, respectively. The mice in NDMA and Cs plus NDMA groups received NDMA at a concentration of 12.5 ppm in drinking water for 8 weeks ad libitum. Without receiving any further treatment, mice were fed the appropriate amount of sterilized commercial diet and water ad libitum. The mice from each group were euthanized and sacrificed sequentially after 1, 3, and 6 months.

### 2.6. Subcellular Fractionation 

The isolation of cytosolic and membranous proteins was carried out using the Plasma Membrane Protein Extraction Kit (Abcam) following the manufacturer’s instructions. The protein concentration of extracts was measured and stored in aliquots at −80 °C until use.

### 2.7. Immunoblot Analysis

Protein concentrations from cytosolic and membranous fractions were measured using the BCA Protein Assay kit (Thermo-Scientific, Waltham, MA, USA). Twenty micrograms of cytosolic and membranous proteins were separated by 12% SDS-PAGE followed by transfer to nitrocellulose membranes (GE Healthcare Biosciences, Uppsala, Sweden). The membranes were blocked for 1 h with 5% skim milk in TBS/Tween 20 (TBST; 10 mM Tris-HCl, pH 7.4, 100 mM NaCl, and 0.1% Tween 20). The membranes were then probed with primary antibodies specific to target proteins. After incubating with the appropriate HRP-conjugated secondary antibodies, the immunoreaction was detected using enhanced chemiluminescence (ECL; Dong-in LS, Seoul, Korea) and quantitated with an Image-Quant LAS 500 biomolecular imager (GE Healthcare Biosciences). Blots were normalized for protein loading with antibodies against GAPDH (for cytosolic proteins) or calnexin (for membranous proteins).

### 2.8. Detection of Intracellular ROS 

The measurement of intracellular ROS levels was performed using a fluorescent dye, 5-(and-6)-chloromethyl-2′,7′-dichlorodihydrofluorescein diacetate acetyl ester (CM-H_2_DCFDA; Molecular Probes, Inc., Eugene, OR, USA). Cells seeding on 96-well black plates at a density of 8 × 10^3^ cells/well were incubated in FBS-free medium containing ESPs, NDMA, or ESPs plus NDMA for 24 h without or with DPI or NAC pretreatment as described above. Cells were washed once with Hank’s balanced salt solution (HBSS), followed by incubation with 5 μM CM-H_2_DCFDA for 20 min at 37 °C in the dark. The DCF fluorescence levels were immediately measured using Perkin-Elmer VICTOR 3 luminescence spectrometer (Perkin-Elmer, Waltham, MA, USA) with excitation and emission wavelengths of 485 and 535 nm, respectively, after washing twice with HBSS. The values were converted to folds for comparison with the untreated control.

### 2.9. Immunohistochemistry

The livers from mice in 4 groups (Con, Cs, NDMA, Cs plus NDMA) were trimmed and fixed in 10% neutrally buffered formalin (pH 7.0). These fixed livers were embedded in paraffin wax and solidified on ice. Paraffin-embedded tissue sections (5 μm thick) were deparaffinized in xylene, rehydrated in a graded ethanol series, and rinsed with distilled water. The sections were heated in a microwave oven in 10 mM sodium citrate buffer (pH 6.0; Vector Laboratories, Burlingame, CA, USA) for antigen retrieval and then incubated in 3% H_2_O_2_/methanol solution to inactivate endogenous peroxidase activity. The sections were incubated with TBST containing 5% normal goat serum for 20 min and the primary antibody (1:100 dilution for p47^phox^ and fibronectin; 1:50 for p67^phox^) overnight at 4 °C. Nonimmune serum with the same dilution factor was substituted for the primary antibody as a control for nonspecific binding. The sections were then incubated with host-specific secondary antibodies (1:200 dilution) for 30 min, followed by treatment with the ABC-HRP kit (VECTASTAIN^®^, Vector Laboratories). Immunostaining of each protein was visualized with the DAB Peroxidase Substrate kit (Vector Laboratories). The sections were briefly counterstained with Harris hematoxylin (Sigma-Aldrich). For visualization of specific collagen fiber types I and III, deparaffinized and rehydrated slides were stained with Harris hematoxylin and rinsed with distilled water for 5 min. Sirius red solution (Abcam) was then applied for 1 h, followed by washing with 0.5% acetic acid solution. Air-dried slides were sealed with organic mounting medium and images were photographed with an upright microscope (Nikon Eclipse Ci, Tokyo, Japan).

### 2.10. Statistical Analysis

Data were expressed as means ± standard error (SE) of three or more independent experiments. Statistical differences among the groups were evaluated by one-way or two-way analysis of variance (ANOVA), followed by Student’s *t*-test or post-hoc Tukey test, as appropriate, using SigmaPlot software (version 8.0; Jandel Scientific., San Rafael, CA, USA). Statistical significance was accepted at *p* < 0.05.

## 3. Results

### 3.1. Activation of NOX and Increased Expression of Fibrosis-Related Proteins in ESPs-, NDMA-, or ESPs plus NDMA-Treated H69 Cells

The exposure of human CCA cells (HuCCT1) to *C. sinensis* ESPs was previously reported to trigger enzymatic ROS generation [17]. Furthermore, ROS generation is well-established to serve as mediator of molecular events during fibrogenesis [2]. These prompted the examination of NOX activation in normal human cholangiocytes, which was one of the major ROS-producing enzymes in response to ESPs, along with the NDMA carcinogenic effect. NOX activation was assessed by determining the accumulation of its subunits (p47^phox^ and p67^phox^) in the cell membrane. Figure 1A shows that the levels of p47^phox^ and p67^phox^ proteins were significantly increased in membrane fraction following 24-hour treatment with ESPs, NDMA, or ESPs plus NDMA. This result indicated that ESPs and NDMA treatment promoted the translocation of NOX subunits from the cytosol to the membrane. This activation subsequently resulted in significant increases in intracellular ROS generation accessed by ROS fluorescent probe, CM-H_2_DCFDA (Figure 1B). The expression of fibrosis marker proteins (e.g., fibronectin and collagen type I) were then evaluated to examine the involvement of NOX activation in fibrogenesis. Immunoblot analyses showed that the levels of fibronectin and collagen type I expression were significantly increased in ESPs-, NDMA-, or ESPs plus NDMA-treated cells by ~1.9-, ~2.7-, or ~3.0-fold over the control level for fibronectin and ~1.8-, ~2.9-, or ~3.6-fold over for collagen type I, respectively (Figure 1C). Taken together, these results indicated that the combination of ESPs and NDMA were synergistic for the generation of ROS and the expression of fibrosis-related proteins.

### 3.2. Inhibitory Effect of NOX Inactivation on the Expression of Fibrosis-Related Proteins in ESPs-, NDMA-, or ESPs plus NDMA-Treated H69 Cells

Cells were pretreated with 20 μM DPI (a NOX inhibitor) or 0.01% DMSO (vehicle only; untreated control) for 1 h, followed by exposure to ESPs, NDMA, or ESPs plus NDMA for 24 h to confirm that NOX activation induced by ESPs or/and NDMA was responsible for the induction of fibrosis-related protein expression. Figure 2A shows that immunoblots depicted that pretreatment with DPI significantly inhibited the accumulation of both membranous p47^phox^ and p67^phox^ induced by each or both (~3.6-, ~1.9-, or 1.8-fold decreases for p47^phox^ and ~1.5-, ~2.5-, or ~2.6-fold decreases for p67^phox^ expression, compared with untreated groups). Concomitant reduction of ROS generation was obvious in these pretreated groups (Figure 2B), indicating that NOX was an indispensable enzyme for ROS generation in ESPs, NDMA, or ESPs plus NDMA-treated cells. Finally, we examined the effect of NOX inhibition on the expression of fibronectin and collagen type I, as accessed by immunoblot analysis. Figure 2C shows that pretreatment with DPI resulted in ~1.7-, ~1.8-, or ~2.6-fold reduction of fibronectin expression in ESPs, NDMA, or ESPs plus NDMA-treated cells, respectively, compared with that of the untreated control group. In addition, DPI pretreatment significantly decreased the levels of collagen type I expression by ~1.5-, ~3.2-, or ~2.6-folds in ESP-, NDMA-, or ESPs plus NDMA-treated cells, respectively, relative to that of the untreated control group. However, these reduced expressions were significantly higher compared with the untreated control, suggesting that additional enzyme(s) may be involved in the induction of fibronectin and collagen type I expression.

### 3.3. Effect of ROS Scavenger on NOX Activation and of Fibrosis-Related Protein Expression

Cells were pretreated with an equal volume of H_2_O (vehicle only; untreated control) or 1 mM NAC, an ROS scavenger, for 1 h, and then exposed to ESPs, NDMA, or ESPs plus NDMA for another 24 h to validate that ROS are responsible for ESP- and/or NDMA-induced fibrosis. Figure 3A shows that NAC pretreatment significantly attenuated the levels of intracellular ROS production triggered by ESPs and NDMA compared with that of the vehicle only control. This pretreatment did not affect the levels of p47^phox^ and p67^phox^ accumulation in the membrane as accessed by immunoblot analysis (Figure 3B), indicating that NAC pretreatment did not exert influence on NOX activation but its outcomes. Moreover, the changes in the levels of fibrosis-related protein expression under reduced ROS conditions were further examined. Figure 3C shows that NAC pretreatment reduced the levels of fibronectin expression by ~2.5-, ~2.0-, or ~3.3-folds induced by ESPs, NDMA, or ESPs plus NDMA, respectively, relative to that of the vehicle only control. In addition, NAC pretreatment similarly decreased the induction of collagen type I expression by ~3.0-, ~2.6-, or ~3.2-folds compared with the vehicle only control. This result indicated that the levels of fibrosis-related protein expression were influenced by the amount of ROS production.

### 3.4. Expression of NOX Subunits (p47^phox^ and p67^phox^) in Mouse Livers Infected with C. sinensis or/and NDMA Uptake

The in vivo relationship of NOX activation with fibrosis in the livers of *C.*
*sinensis-*infected and NDMA-administered mice was examined next. Mice from four groups (Con, Cs, NDMA, Cs plus NDMA) were necropsied at 1, 3, and 6 months p.i., and liver tissues were then collected for immunohistochemical analyses. Histopathological changes such as hyperplasia and fibrosis of biliary epithelia in the liver sections of Cs and Cs plus NDMA groups were evident between 1 and 6 months. However, no obvious changes were detectable in those of Con and NDMA groups at the entire experimental periods (Figure 4). Bile ducts of Cs and Cs plus NDMA groups were densely filled with brownish particles, which may be an admixture of worm eggs, mast cells, and biliary sludge. Consequently, immunohistochemical expression of both p47^phox^ and p67^phox^ were hardly detectable in 1- to 6- (Con) and 1- to 3- (NDMA) month-old mice livers. Their immunoreactivities were significantly increased in the 6-month-old (NDMA) livers, even though no obvious histopathological changes were observed. As early as 3 months, p.i., p47^phox^, and p67^phox^ immunoreactivities were observed in the fibrotic regions of Cs livers. These reactivities were more intense at 6 months. Their expressions in Cs plus NDMA livers were observed in both inflammatory and fibrotic regions at 1 month. These expressions were strong and extended to the hyperplasia region between 3 and 6 months. These results indicated that *C. sinensis* infection and NDMA administration synergistically activated NOX at the late stage of infection, which contributed to increases in persistent ROS accumulation in the liver tissues.

### 3.5. Collagen and Fibronectin Expression in Mouse Livers Infected with C. sinensis or/and NDMA Uptake

The deposition of collagen fibers and the fibronectin expression in the liver tissues of four groups (Con, Cs, NDMA, Cs plus NDMA) were examined because hepatic fibrosis was one of the most common pathological symptoms provoked by *C. sinensis* infection and NDMA treatment. Figure 5A shows that Sirius red staining exhibited minimal collagen deposition around the biliary triad in Con group during the entire experimental period. These collagen fibers were densely deposited around the bile ducts and blood vessels in the livers of Cs and NDMA groups as early as 1 month after infection or administration and became denser between 3 and 6 months (Figure 5A,C). Dense collagen fibers were deposited in the bile duct epithelial and hyperplasia regions at 1 month and extended to fibrous stroma formation in Cs plus NDMA livers. Fibronectin immunoreactivity in Con livers was hardly detectable except for bile duct epithelia during the entire experimental periods, while its reactivity was detectable in Cs and Cs plus NDMA groups at 1 month. Reactivity dramatically increased in NDMA groups at 6 months (Figure 5B,D). In particular, the intensity and immunoreactivity became stronger and broader in proportion to the infection and treatment duration. Taken together, these results indicated that the expression patterns of collagen and fibronectin were similar to those of p47^phox^ and p67^phox^.

## 4. Discussion

Liver fluke infection is a well-known major cause of several hepatobiliary disorders including chronic inflammation and periductal fibrosis via worms and their ESPs. Excessive generation of free radicals contributes to the promotion of these pathologic events, with tight interdependent relationships among oxidative stress, inflammation, and fibrosis. These are key pathologic factors in liver fluke-associated CCA animal models, along with certain carcinogens including nitrosoamine compounds, which may be obtained from endogenous sources or parasites [9,10]. Free radicals enzymatically triggered by *C. sinensis* ESPs were previously reported to exert NF-κB-mediated inflammation in CCA cells [17]. Moreover, periductal fibrosis, inflammatory infiltration, and oxidative stress markers were increased in the livers of *C. sinensis*-infected mice, accompanied by differential elevations of proinflammatory cytokine levels in their sera [12]. The present study examined that ROS generation induced by the activation of NOX promoted biliary fibrosis in ESPs- and NDMA-exposed H69 cells. Furthermore, the differential expression of NOX subunits and fibrosis markers in the livers of mice infected with *C. sinensis* or/and administered with NDMA were profiled.

Overactive NOXs are associated with an oxidative cell or tissue injury, which causes the initiation and progression of various vascular disorders and amylotropic lateral sclerosis, as well as liver inflammation and fibrosis [16]. Angiotensin II (Ang II) and phorbol ester (e.g., phorbol myristic acetate) are commonly used activators of NOX in vitro study whose treatment activated NOX type 2 in pulmonary microendothelial cells, alveolar macrophages, and polymorphonuclear leukocytes, resulting in increased superoxide/H_2_O_2_ generation [20]. Hepatocyte NOX enzymes were a prominently endogenous source of ROS and peroxynitrite generation in hepatitis C virus-infected human liver [21]. *C. sinensis* ESP treatment resulted in the translocation of p47^phox^ and p67^phox^ to plasma membrane in normal cholangiocytes (H69 cells), along with an increase in ROS generation (Figure 1A,B), similar to previous findings in CCA cells [17]. This result indicates that NOX activation occurs in both cancerous and noncancerous bile duct epithelial cells in response to ESPs. Moreover, the accumulation of these NOX subunits in the plasma membrane was observed in NDMA-exposed cells, suggesting that NDMA may function as a promoter for NOX activation. However, the exact molecular mechanism of NDMA-mediated NOX activation remains elusive. One possible mechanism would be ROS-triggered transcriptional regulation of NOX subunits, in which free radicals are generated during enzymatic and nonenzymatic degradations of NDMA [22]. Furthermore, NDMA decreased the activities of antioxidant enzymes such as superoxide dismutase, catalase, glutathione peroxidase (GPx), and glutathione S-transferase (GST), resulting in redox imbalance [23]. NF-κB, activator protein 1, and CCAAT/enhancer-binding protein are redox-active transcription factors that are present at the promoter regions of genes encoding NOX subunits, demonstrating their essential roles in NOX activity [24,25,26]. Therefore, speculating that these ROS-induced transcription factors may be coordinately involved in the upregulation of NOX subunit expression in NDMA-treated H69 cells is plausible. In addition, NDMA ingestion or administration triggers the immune system, subsequently producing various proinflammatory cytokines such as IL-β1, IL-6, IL-22, IFN-γ, and TNF-α. These cytokines activate downstream NF-κB- and transforming growth factor-β (TGF-β)-mediated signaling pathways [22], which in turn may induce NOX activation. Treatment with ESPs and NDMA synergistically increased ROS production (Figure 1B). This synergistic effect of *C. sinensis* ESPs and NDMA was also reported in human embryonic kidney cells (HEK293T) and H69 cells, where their co-treatment induced cell proliferation and gap–junction proteins connexin 43 and 26 expression, respectively [27,28].

Chronic oxidative stress is well-known to play pivotal roles on the initiation and progression of liver fibrosis, wherein ROS stimulates the production of profibrogenic mediators from Kupffer cells, thus circulating inflammatory cells, and directly activate HSCs and induce transformation into myofibroblast-like cells. In particular, the NOX isoforms are expressed in different types of liver-resident cells, thus NOX-derived ROS is essential for the pathogenesis of liver fibrosis, together with the cytokine–receptor interactions [29,30]. The fibrotic process is characterized by the induction of fibrogenic cytokines, increased deposition, and altered ECM composition, which includes the changes in different collagen types and increases in the expression of TGF-β, α-smooth muscle actin (α-SMA), and ECM components (e.g., fibronectin and laminin) [29]. Concerning liver fluke infestation, elevated levels of collagen type I, TGF-β1, and α-SMA mRNA were reported to be proportional to the degree of hepatic fibrosis in *C. sinensis*-infected mice, which was mediated by TGF-β/Smad signaling pathway [31]. ECM analysis and adhesion molecule polymerase chain reaction array in *C. sinensis* ESP-treated HuCCT1 cells revealed increases in collagen 1α subtypes and fibronectin transcriptions, together with those of membrane-associated MMPs [32]. The difference in collagen metabolism mediated by differential expression of MMPs and tissue inhibitors of MMPs would affect *O. viverrini* infection-induced peribiliary fibrosis and liver injury [14]. The levels of fibronectin and collagen I expression in the present study were elevated in the ESP-, NDMA-, or ESPs plus NDMA-treated H69 cells (Figure 1C). Consistent with this result, collagen synthesis and deposition of collagen, laminin, and fibronectin increased in NDMA-induced hepatic fibrosis rat model [33]. Taken together, both ESPs and NDMA promote biliary fibrosis and synergistically function as fibrotic mediators in cholangiocytes.

Excessive ROS formation generated by NOX complex dysregulation has been well established as one of the critical factors for fibrogenesis. For example, bile duct ligation-induced liver injury and fibrosis were significantly attenuated in p47^phox^ knockout mice, concomitant with the reduced expression of α-SMA and TGF-β compared with those of the wild-type counterparts [34]. Hepatic fibrosis and ROS generation were attenuated in both NOX1 and NOX2 (gp91^phox^) null mice after CCl_4_ injection or bile duct ligation. In addition, both NOX1- and NOX2-deficient HSCs exhibited the deterioration of ROS generation and collagen I/TGF-β upregulation in response to Ang II [16]. Subcutaneous injection of DPI effectively prevented alcohol-induced liver damage in rats by inhibition of NOX-dependent free radical generation, thus attenuating NF-κB activation and induction of TNF-α mRNA expression [35]. Moreover, the present study showed that the pretreatment of cells with DPI decreased NOX activation triggered by ESPs or/and NDMA and subsequent correlation reduced intracellular ROS levels (Figure 2A,B). The DPI inhibitory effect on NOX-mediated ROS production was also reported in growth factor-induced human keratinocyte (HaCaT) migration [36] and a skin fibrosis model induced by TGF-β [37]. NOX inhibition was observed to lead to less accumulation of fibrotic markers, including fibronectin and collagen type I, than those of DPI-untreated cells (Figure 2C), indicating that NOX mediates the actions of *C. sinensis* ESPs and NDMA and plays a critical role in biliary fibrogenesis. In accordance with this finding, the inhibition of NOX activity by DPI has been reported to prevent the induction of collagen I, α-SMA, and fibronectin genes in the bleomycin-induced scleroderma mouse model, attenuating skin fibrosis and myofibroblast activation [37].

ROS detoxification can alleviate liver fibrosis progression given the detrimental role of NOX-dependent ROS in liver fibrogenesis. For example, NAC is a thiol-containing antioxidant that facilitates reduced glutathione (GSH) biosynthesis and scavenges ROS formed during oxidative stress. Mice intraperitoneally injected with NAC prior to hepatic ischemia-reperfusion showed that ROS-induced liver injury was significantly attenuated via the inhibition of endoplasmic reticulum stress and apoptosis [38]. NAC administration in a CCl_4_-inhaled cirrhotic rat model resulted in a lesser decrease in GPx levels and collagen deposition with a lower degree of fibrosis than those of only CCl_4_-treated counterparts [39]. Post-treatment with NAC protected rat livers from NDMA-induced oxidative stress and hepatocellular damage, restoring the activities of both the enzymatic and nonenzymatic antioxidants [40]. Restoring GSH levels in the livers of *O. felineus*-infected hamsters under NAC uptake in drinking water allowed the activation of GPx and GST as well as decreases in collagen content and liver fibrosis severity [41]. Pretreatment with NAC attenuated NF-κB-mediated inflammation in HuCCT1 cells exposed to *C. sinensis* ESPs [17]. The present study showed that the pretreatment of H69 cells with NAC reduced the levels of fibronectin and collagen type I expression induced by ESPs or/and NDMA without significantly affecting the translocation of NOX subunits to the plasma membrane (Figure 3), implying that NAC may not participate in NOX activity inhibition. The NAC pretreatment effect on the modulation of fibrotic marker protein expression was also reported in rat renal fibroblast cells in response to Ang II [42]. Taken together, these findings suggest that an antifibrotic property of an antioxidant such as NAC effectively prevents the progression of various tissue fibrosis through the maintenance of redox homeostasis. Persistent liver fluke infection causes oxidative insult-mediated liver injuries including inflammation and periductal fibrosis, promoting the more severe chronic hepatobiliary abnormalities and even CCA [5,7,43]. Immunoreactivities of both p47^phox^ and p67^phox^ in the present study were observed in inflammatory and fibrotic regions of mouse livers infected with *C. sinensis* (Figure 4). These immunostained regions coincide with accumulated regions of lipid peroxidation products and oxidative DNA adducts in infected mouse livers [12], suggesting that excessive ROS generated by NOX overactivation may contribute to the induction of oxidative stress markers. Their intensities were proportional to the lesion severities caused by infection duration. No changes were noted in the expression of NOX subunits until the 3-month NDMA group. However, their increases were noted in the 6-month-treated group compared with those of the controls. In addition, no obvious histopathological changes were observed in this group during the entire experimental period. This may be due to insufficient NDMA dosage or/and its inconsistent uptake through drinking water. The same NDMA subcarcinogenic dose has also been reported in *C. sinensis*-associated CCA animal models in which NDMA administration alone did not cause any obvious histopathological changes in the livers of hamsters and mice [13,44]. Meanwhile, lesion severity and the levels of p47^phox^ and p67^phox^ immunoreactivities in livers of the *C. sinensis* plus NDMA-infected group were higher compared with the *C. sinensis*-infected group alone, indicating that they synergistically act on the liver deterioration as well as NOX activation. It is believed that this is the first report of p47^phox^ and p67^phox^ immunoreactivities to mouse liver tissues infected with *C. sinensis* and/or administered NDMA. The brownish pigments inside the bile ducts observed in both *C. sinensis* and *C. sinensis* plus NDMA groups (Figure 4) may be admixtures of mast cells and *C. sinensis* eggs adhered to or wrapped by bilirubinate granules, mucoid matter, and calcium carbonate crystals involved in the formation of gallbladder stones, as described in cholecystolithiasis patients associated with *C. sinensis* infection [45].

Inadequate accumulation of ECM proteins is a common pathologic step in liver fibrosis, leading to disruption of tissue microarchitecture and liver dysfunction [11]. Liver fibrosis induced by CCl_4_- or NDMA-treated mice exhibited increases in the amount of both fibronectin and collagen in ECM. Blockade of fibronectin deposition via a specific peptide diminished fibronectin and collagen type I protein accumulation in the matrix, improving liver function during fibrogenesis [46]. The presence of collagen fibers in and around the bile duct in the present study was observed in Cs, NDMA, and Cs plus NDMA groups. In particular, these fibers were the most dense and extensive in both treated groups (Figure 5A), indicating that *C. sinensis* infection and NDMA treatment synergistically foster deposition and synthesis of hepatic ECM components including various collagen subtypes, consequently leading to biliary fibrosis. Differential overexpression of collagen types I and IV in a *C. sinensis*-associated CCA animal model has suggested that type I is involved in several signaling pathways of cholangiocarcinogenesis, while type IV is for the advancement of malignant CCA [13]. Immunostaining of fibronectin was detected in Cs, NDMA, and Cs plus NDMA groups following collagen accumulation, and its intensity was increased in a time-dependent manner (Figure 5B). Fibronectin dysregulation is an early and progressive event during liver fibrogenesis in vivo and in vitro, wherein fibronectin fibril formation is required for collagen matrix deposition [47]. The immunoreactivity of α-SMA was detectable in the livers of each group with similar expression patterns of collagen fibers and fibronectin (data not shown) as reported in the fibrotic livers of *O. viverrini*-infected hamsters [48]. Moreover, increased accumulation of collagen fibers, fibronectin, and α-SMA in the livers of the NDMA group without marked visual lesions were found, implying that dysregulation of ECM components biochemically proceeds phenotypical damage in this group. These findings suggest that excessive accumulation and turnover of ECM components modulate cell survival, proliferation, interaction with cell adhesion molecules, migration, and gene expression, consequently leading to malignant transformation, given the close relationship of oxidative stress with fibrosis.

This study recognized that tumor onset and progression were not evident in a murine model during the entire experimental periods unlike clonorchiasis-associated CCA development in hamsters [13]. Nevertheless, NOX-mediated oxidative insults were found to stimulate the biliary fibrosis in *C. sinensis* infective in vitro and in vivo models, which may function as a detrimental promoter for the development of more severe biliary disorders. Consistent with this, the histopathologic changes (e.g., hyperplasia, peliosis, and extensive biliary fibrosis) in the livers of Cs plus NDMA mice were found. However, no CCA development was observed [44].

## 5. Conclusions

In conclusion, this study showed that treatment of H69 cells with *C. sinensis* ESPs or/and NDMA triggers NOX-mediated oxidative insult and subsequent increases in collagen type I and fibronectin expression. The intensity and extent of immunoreactive NOX subunits and the deposition of collagen fibers and fibronectin staining in infected/administrated livers are proportional to fibrotic severity and the duration, indicating that constitutive NOX overactivation contributes to biliary fibrosis. Oxidative stress-mediated biliary fibrosis creates a vulnerable microenvironment that may promote malignant transformation for the development of advanced hepatobiliary diseases including CCA, even though no carcinogenesis is evident in the clonorchiasis plus NDMA-associated murine model. These findings broaden the understanding of the progression of oxidative stress-induced biliary fibrosis during *C. sinensis* infection or/and NDMA administration and provide a new basis for therapeutic approaches to control clonorchiasis-associated biliary fibrosis.

## Figures and Tables

**Figure 1 antioxidants-10-00869-f001:**
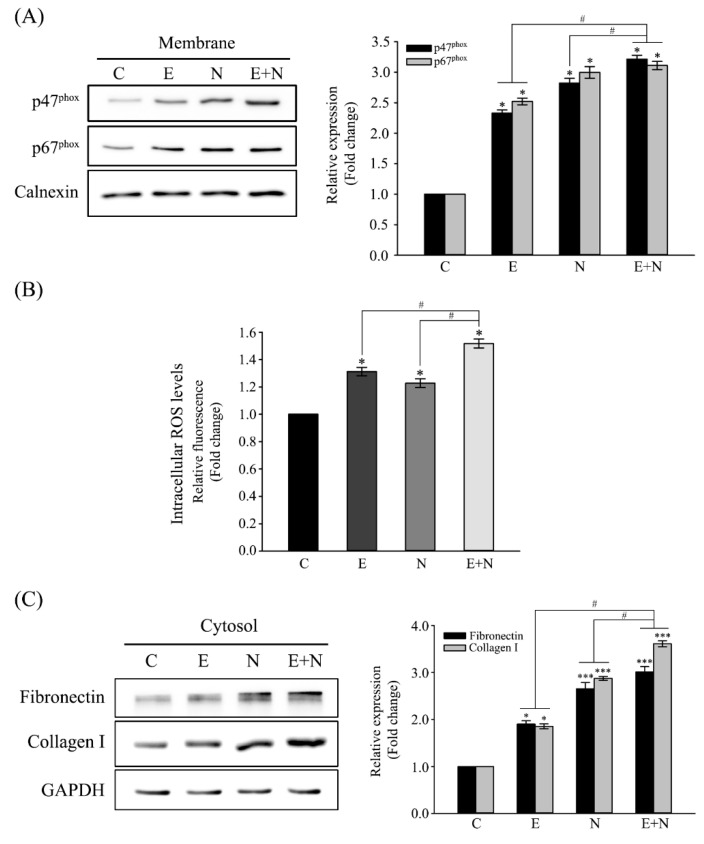
Effect of ESPs or/and NDMA on NOX activation, ROS generation, and fibrosis-related protein expression. Cells were treated with 1.6 μg/mL ESPs, 8 μg/mL NDMA, or both for 24 h and harvested for immunoblot and DCF analyses. (**A**) Representative immunoblot showing the accumulation of the NOX subunits (p47^phox^ and p67^phox^) in the membrane fractions. Individual bands were quantified densitometrically and normalized to calnexin. Values are presented as means ± SE for three independent experiments. *^, #^
*p* < 0.05, *; compared with the untreated control, ^#^; ESPs- or NDMA-treated group versus ESP and NDMA-treated group. (**B**) Intracellular ROS levels were measured by CM-H_2_DCFDA fluorescence with a spectrofluorometer. Each data set represents the mean ± SE (*n* = four independent experiments) expressed as a fold change of untreated control value. *^, #^
*p* < 0.05, *; compared with the untreated control, ^#^; ESP- or NDMA-treated group versus ESP and NDMA-treated group. (**C**) Representative immunoblot showing fibrosis-related proteins in cytosolic fractions. Protein bands were quantified using densitometry, and their abundances were normalized according to GAPDH expression. Each data set represents the means ± SE (*n* = three independent experiments). *^, #^ *p* < 0.05, *** *p* < 0.001, *; compared with the untreated control, ^#^; ESP- or NDMA-treated group versus ESPs and NDMA-treated group. C; untreated control, E; ESP-treated, N; NDMA-treated, E + N; ESPs and NDMA-treated.

**Figure 2 antioxidants-10-00869-f002:**
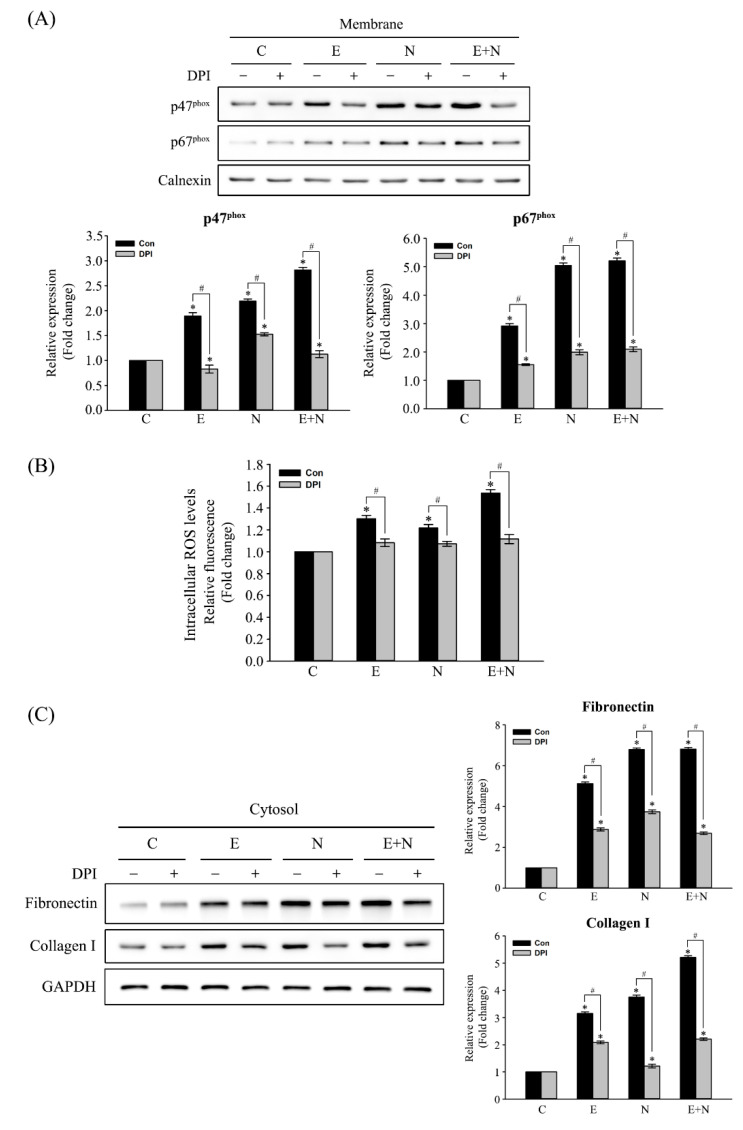
Inhibitory effect of NOX inhibitor on ESPs and/or NDMA-induced NOX activation, ROS generation, and fibrosis-related protein expression. Cells were pretreated with 20 μM DPI or 0.01% DMSO (vehicle only; untreated) for 1 h and then treated with ESPs, NDMA, or ESPs plus NDMA for 24 h. (**A**) Membrane proteins were immunoblotted for p47^phox^ and p67^phox^. Calnexin was used as a protein loading control. *^, #^
*p* < 0.05, *; compared with the untreated control, ^#^; vehicle only versus DPI-treated group. (**B**) The effect of DPI pretreatment on ROS generation triggered by ESPs and/or NDMA. Each data set represents the means ± SE (*n* = four independent experiments) expressed as a fold change of untreated control. *^, #^ *p* < 0.05, *; compared with the untreated control, ^#^; vehicle only versus DPI-treated group. (**C**) Expression of fibrosis-related proteins. GAPDH was used as a loading control for normalization. Data in graphs in (**A**,**C**)a re expressed as means ± SE (*n* = three independent experiments). *^, #^
*p* < 0.05, *; compared with the untreated control, ^#^; vehicle only versus DPI-treated group.

**Figure 3 antioxidants-10-00869-f003:**
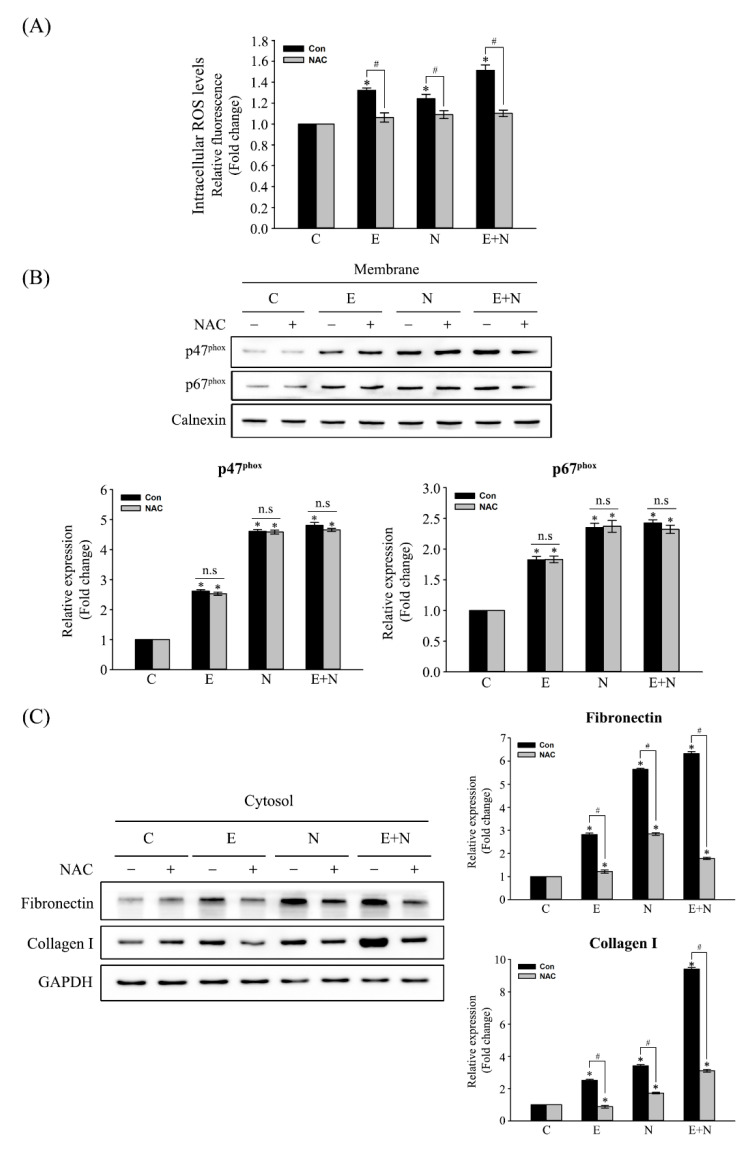
Effects of NAC on ESPs and/or NDMA-induced NOX activation, ROS generation, and fibrosis-related protein expression. Cells were pretreated with 1 mM NAC or an equal volume of H_2_O (untreated) for 1 h, followed by the treatment with ESPs, NDMA, or ESPs plus NDMA for 24 h. (**A**) The levels of ROS determined by DCF fluorescence. Each data set represents the means ± SE (*n* = four independent experiments), expressed as a fold change of control values. *^, #^
*p* < 0.05, *; compared with the untreated control, ^#^; vehicle only versus NAC-treated group. (**B**) Representative immunoblots for membranous p47^phox^ and p67^phox^ expression. (**C**) The inhibitory effect of NAC on ESPs and/or NDMA-induced fibrosis-related protein expression. Individual bands were quantified densitometrically and normalized to calnexin and GAPDH as loading controls for membrane (**B**) and cytosolic (**C**) proteins. The values in graphs in **B** and **C** are represented as fold changes relative to the control expressed as means ± SE (*n* = three independent experiments). *^, #^ *p* < 0.05, *; compared with the untreated control, ^#^; vehicle only versus NAC-treated group, n.s; not significant.

**Figure 4 antioxidants-10-00869-f004:**
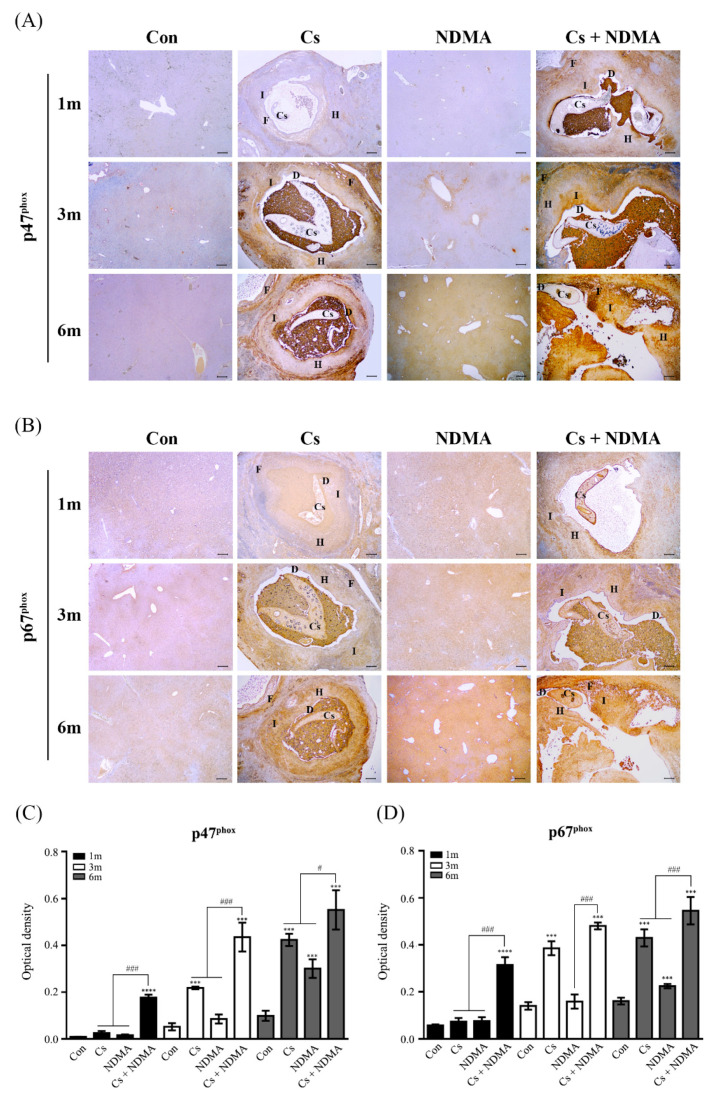
Time course of p47^phox^ and p67^phox^ immunoreactivity in the livers of Con (uninfected control), Cs (*C. sinensis* infected), NDMA (NDMA administered), and Cs plus NDMA (*C. sinensis-*infected and NDMA-administered) mice. Three independent liver sections from each group for three infection time points were immunostained with p47^phox^ or p67^phox^, and representative images for each group are shown in (**A**,**B**), respectively. p47^phox^ and p67^phox^ expression was marked with brown deposits and nuclei were counterstained with hematoxylin (*blue*). Scale bar = 200 μm; original magnification, ×40. (**C**,**D**) The intensities of dark brown dots in (**A**,**B**) images were quantified using an Image J program. Each data set in the graph is expressed as means ± SE. ^#^ *p* < 0.05, ***, ^###^ *p* < 0.001, **** *p* < 0.0001, compared with the control group, ^#^; Cs or NDMA group versus Cs plus NDMA group. Cs, *C. sinensis* worm; D, ductal dilatation; F, Fibrosis; H, hyperplasia region; I, inflammatory region.

**Figure 5 antioxidants-10-00869-f005:**
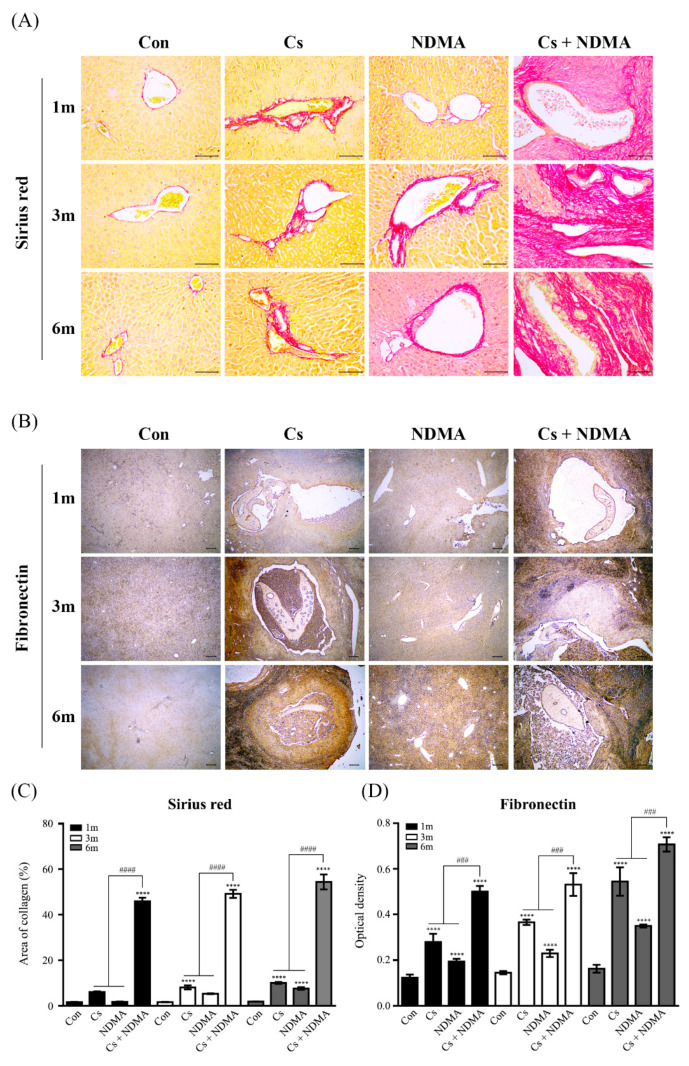
Time profiles of collagen and fibronectin expression in the livers of mice from 4 groups (Con, Cs, NDMA, Cs plus NDMA). Three independent liver sections from each group for three infection time points were stained with Sirius red for collagen types I and III fibers or immunostained with polyclonal antibody to fibronectin, and representative images for each group are shown in (**A**,**B**), respectively. Collagen fiber deposition is presented in red. Brown represents the expression of fibronectin and blue for counterstained nuclei with hematoxylin. (**A**) Scale bar = 100 μm. Original magnification, ×200. (**B**) Scale bar = 200 μm; original magnification ×40. (**C**) Quantification of the Sirius red staining area using an Image J program. (**D**) The intensities of dark brown dots in the B image were quantified using an Image J program. Data in the graph are expressed as means ± SE. ^###^ *p* < 0.001, ****^, ####^ *p* < 0.0001, *; compared with the control group, ^#^; Cs or NDMA group versus Cs plus NDMA group.

## Data Availability

All data is contained within the article.

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
