# Peer review of "The Overactivation of NADPH Oxidase during Clonorchis sinensis Infection and the Exposure to N-Nitroso Compounds Promote Periductal Fibrosis"

_antioxidants, 2021, doi:10.3390/antiox10060869_

Round 1
Reviewer 1 Report
It is necessary to summarize the contents of the paper more.
Author Response
It is necessary to summarize the contents of the paper more.
: The manuscript has substantially been modified to re-summarize the contents, which include rephrasing the title and overlapped sentences, rewriting the abstract.
Reviewer 2 Report
One of my major concerns is the statistical analysis of the data. The only thing the authors do is to use means and s.e. and compare treatments to controls using a t-test. However, a t-test requires normally distributed data and the data should be tested for this. Otherwise, non-parametric tests have to be applied. Furthermore, the experimental design is actually 2x2 design and should be analysed in this way, showing what the effect of ESP and NMDA alone is and how it acts in combination. If data are normally distributed this can be done via an ANOVA. Experimental replicates can be treated as random factors. This procedure will also avoid the multiple testing problem that currently is not addressed at all. My feeling is that the results presented under 3.3 are not all significant as the results presented in Fig. 3b are so close to each other with overlapping s.e. that either they can’t be significant or after controlling for multiple testing it will not any longer be significant. I suggest to use appropriate statistical testing accounting for multiple testing (e.g. Bonferroni correction) or use test that include all data (e.g. ANOVA or LM/LMM). Technical replicates should be used appropriately and not as biological replicates.
Try to avoid expressions when referring to the gene or protein expression, use just the singular, e.g. line 104 or line 107)
Line 228 mediator instead of mediators
Line 403 in vitro in italics
Author Response
Reviewer 2>
- One of my major concerns is the statistical analysis of the data. The only thing the authors do is to use means and s.e. and compare treatments to controls using a t-test. However, a t-test requires normally distributed data and the data should be tested for this. Otherwise, non-parametric tests have to be applied. Furthermore, the experimental design is actually 2x2 design and should be analysed in this way, showing what the effect of ESP and NMDA alone is and how it acts in combination. If data are normally distributed this can be done via an ANOVA. Experimental replicates can be treated as random factors. This procedure will also avoid the multiple testing problem that currently is not addressed at all. My feeling is that the results presented under 3.3 are not all significant as the results presented in Fig. 3b are so close to each other with overlapping s.e. that either they can’t be significant or after controlling for multiple testing it will not any longer be significant. I suggest to use appropriate statistical testing accounting for multiple testing (e.g. Bonferroni correction) or use test that include all data (e.g. ANOVA or LM/LMM). Technical replicates should be used appropriately and not as biological replicates.
: ANOVA had been initially used for statistical differences among the four groups (e.g. Con, ESP, NDMA, and ESP + NDMA). In addition to Students’ t-test, post-hoc Tukey’s test had been used for the comparison between the groups such as Con vs ESPs, Con vs NDMA, etc. This had been clearly described in the subsection of Materials and Methods (2.10. Statistical analysis; p5, ln234-237). In Fig.3B, NAC pretreatment did not affect the protein expression, which means no statistically significant differences between vehicle-only versus NAC-treated groups. However, the comparison of untreated control with other groups showed significant differences. Asterisk marks represent this matter.
- Try to avoid expressions when referring to the gene or protein expression, use just the singular, e.g. line 104 or line 107)
: As suggested, all plural words ‘expressions’ presented in the text have been replaced with the singular words ‘expression’.
- Line 228 mediator instead of mediators
: As suggested, ‘mediators’ has been replaced with the ‘mediator’ (p5, ln243).
- Line 403 in vitro in italics
: As suggested, we have corrected it (p15, ln426-427).
Reviewer 3 Report
The Authors have produced a well-structured and articulated paper,
comprehensively analyzing C. sinen-22 sis infection-associated fibrosis.
The revision of the existing literature is broad and up-to-date.
In my opinion the figures could be improved by adding titles above or directly on the y axes of the histograms to make the understanding of the analyzed parameter more immediate. For example in Figure 1B I would add "intracellular ROS levels".
In addition, the manuscript would benefit from a review by a native
English speaker.
The Authors should make minor revisions before this article can be published.
Author Response
The Authors have produced a well-structured and articulated paper, comprehensively analyzing C. sinensis infection-associated fibrosis.
The revision of the existing literature is broad and up-to-date.
- In my opinion the figures could be improved by adding titles above or directly on the y axes of the histograms to make the understanding of the analyzed parameter more immediate. For example in Figure 1B I would add "intracellular ROS levels".
: As suggested, subtitles of parameters have been added to the relevant Y axes of the graphs (Fig. 1B, 2B, and 3A).
- In addition, the manuscript would benefit from a review by a native English speaker.
: This manuscript had been edited for English language, grammar, punctuation, and spelling by an experienced editor whose first language is English and who specializes in editing papers. Certificate of Editing is attached to ‘Cover letter’.
The Authors should make minor revisions before this article can be published.
Reviewer 4 Report
The present manuscript explores the molecular mechanisms leading to liver injury, oxidative stress and subsequent fibrotic changes during Clonorchis sinensis infestation.
The manuscript is presented by experts in the field and represents the logical extension of their previous work. Given the high prevalence of C. sinensis infestations in some world areas, the study is of timely medical relevance. Although the link between C. sinensis infestations and oxidative stress was formerly established by the authors and other groups, the link with NOX and liver fibrosis represents sufficient novelty to warrant publication.
However, the authors should consider the following remarks:
- I have found that the title and the abstract do not reflect the content of the manuscript. The title claims that NDMA leads to periductal fibrosis during C. sinensis infection. Is NDMA produced by C. sinensis? Also, the manuscript provides insufficient evidence for periductal fibrosis being caused by NOX. This would require the use of specific NOX inhibitors in vivo and/or the use of NOX KO mice. Reproducing in mice the experiments with DPI or N-acetylcysteine shown for H69 cells would have provided some evidence on this matter. I would recommend the authors rephrasing the title and re-writing the abstract.
- Authors should make an effort to quantify protein expression and lesion severity in images presented in figures 4 and 5. Also, The figures 4 and 5 legend states that three independent liver sections from each group for "three points" were immunostained. Not clear is what "three points" actually means.
- Methods: please provide the source of H69 cells and shortly explain how C. sinensis excretory-secretory products are obtained, so that the reader is not forced to refer to previous publications.
- What were the vehicles for DPI and NAC? Control cells should not be left untreated, but treated with the vehicle.
Minor comments
Please revise the following phrasing for clarity:
- Line 77: not clear is what "infective duration passage" is.
- Line 94: NOX2 phagocyte > phagocytic (or phagocyte) NOX2
- Figure legend 1: please use the correct abbreviation for DCF, i.e. CM-H2DCFDA
- Line 435: circulate > circulating
Author Response
The present manuscript explores the molecular mechanisms leading to liver injury, oxidative stress and subsequent fibrotic changes during Clonorchis sinensis infestation.
The manuscript is presented by experts in the field and represents the logical extension of their previous work. Given the high prevalence of C. sinensis infestations in some world areas, the study is of timely medical relevance. Although the link between C. sinensis infestations and oxidative stress was formerly established by the authors and other groups, the link with NOX and liver fibrosis represents sufficient novelty to warrant publication.
However, the authors should consider the following remarks:
- I have found that the title and the abstract do not reflect the content of the manuscript. The title claims that NDMA leads to periductal fibrosis during C. sinensisinfection. Is NDMA produced by C. sinensis? Also, the manuscript provides insufficient evidence for periductal fibrosis being caused by NOX. This would require the use of specific NOX inhibitors in vivo and/or the use of NOX KO mice. Reproducing in mice the experiments with DPI or N-acetylcysteine shown for H69 cells would have provided some evidence on this matter. I would recommend the authors rephrasing the title and re-writing the abstract.
: As suggested, we have rephrased the title and abstract. DPI is a well-known inhibitor of NOX that is commonly used for the inhibition of NOX activity in both in vitro cell culture and in vivo animal studies. In the present study, DPI was used to inhibit NOX activity in general, which led to decreases in fibrosis-related protein expressions induced by ESPs or/and NDMA in H69 cells. This provides a sufficient evidence for the involvement of NOX in periductal fibrosis that occurs during the infection. We completely agree to your concern about in vivo DPI or/and NAC administered-mouse studies and the usage of specific NOX subunit KO mice. These studies will strength detrimental role of NOX dysregulation in periductal fibrosis during C. sinensis infection.
- Authors should make an effort to quantify protein expression and lesion severity in images presented in figures 4 and 5. Also, The figures 4 and 5 legend states that three independent liver sections from each group for "three points" were immunostained. Not clear is what "three points" actually means.
: As suggested, we have quantified the immunoreactivies of protein expressions (p47phox, p67phox, and fibronectin) and Sirius red staining regions in control and infected livers in Fig. 4 and 5, using an Image J program. These respective quantifications are presented as the graphs in Fig. 4C, 4D, 5C, and 5D. “Three points” has been replaced with “three infected time points” (p12-14, ln367-368 and 396-397) to clarify the infection times or/and NDMA administration (1,3, and 6 month).
- Methods: please provide the source of H69 cells and shortly explain how C. sinensisexcretory-secretory products are obtained, so that the reader is not forced to refer to previous publications.
: As suggested, the source of H69 cells (p3, ln137-138) and the explanation for ESP preparation have been described in Method section (p3, ln125-134).
- What were the vehicles for DPI and NAC? Control cells should not be left untreated, but treated with the vehicle.
: The vehicles of DPI and NAC are DMSO and H2O, respectively. The ‘DPI- or NAC-untreated group’ has been replaced with the “vehicle only group” (p6-10, ln280-341).
Minor comments
Please revise the following phrasing for clarity:
- Line 77: not clear is what "infective duration passage" is.
: The ‘infective duration passage’ has been replaced with ‘infective periods’ (p2, ln77).
- Line 94: NOX2 phagocyte > phagocytic (or phagocyte) NOX2
: As suggested, ‘NOX2 phagocyte’ has been replaced with ‘phagocytic NOX2’ (p2, ln94).
- Figure legend 1: please use the correct abbreviation for DCF, i.e. CM-H2DCFDA
: As suggested, ‘DCF’ has been replaced with ‘CM-H2DCFDA’ (p6, ln268).
- Line 435: circulate > circulating
: As suggested, ‘circulate’ has been replaced with ‘circulating’ (p15, ln458).
Round 2
Reviewer 1 Report
I think the manuscript has been properly revised.
Author Response
We appreciate your efforts to review this manuscript.
Reviewer 2 Report
I will keep my previous comments alive as these have not been addressed. Your statistical analysis does not reflect the design chosen. You should use proper statistical analysis. Avoid pseudo-replication, avoid multiple testing problems. Test the distribution of data and choose appropriate statistics. Use linear models and factor combinations instead of comparing group 1-4 you should test factor 1, factor 2, and a combination of them. Everythings else does not make sense.
Author Response
I will keep my previous comments alive as these have not been addressed. Your statistical analysis does not reflect the design chosen. You should use proper statistical analysis. Avoid pseudo-replication, avoid multiple testing problems. Test the distribution of data and choose appropriate statistics. Use linear models and factor combinations instead of comparing group 1-4 you should test factor 1, factor 2, and a combination of them. Everythings else does not make sense.
: As the result of using the Shapiro-Wilk test, our data follow normal distribution. Two factors have been considered to statistically analyze the data; the first factor is the ‘Condition’ (e.g. Con, ESP, NDMA, ESP+NDMA) and the second one is ‘Treatment’ (e.g. Vehicle, NAC). We have run a two-way ANOVA, including Tukey test for multiple comparisons using SigmaPlot software. As a result, there are significant differences among ‘Condition’ groups, but no significant differences within ‘Treatment’ or within combination (Condition x Treatment groups). To clarify this matter, the analytical results from Two-way ANOVA and Tukey test used for Figure 3B has been attached below. The figure 3B graph and relevant legends have been accordingly modified (p10, ln341).
Reviewer 4 Report
The Authors fulfilled most of my concerns. I have detected some typos in the revised version, for example, in the title, "promotes" should be "promote"; in the manuscript, "three infected time point" should be "three infection time points". Overall, the manuscript should be submitted to English proofreading prior publication.
Author Response
1. I have detected some typos in the revised version, for example, in the title, "promotes" should be "promote"; in the manuscript, "three infected time point" should be "three infection time points".
: As suggested, typos have been completely corrected.
2.Overall, the manuscript should be submitted to English proofreading prior publication.
: All authors have carefully proofread the revised manuscript.